# Mechanism and Compatibility of Pretreated Lignocellulosic Biomass and Polymeric Mixed Matrix Membranes: A Review

**DOI:** 10.3390/membranes10120370

**Published:** 2020-11-26

**Authors:** Abiodun Abdulhameed Amusa, Abdul Latif Ahmad, Jimoh Kayode Adewole

**Affiliations:** 1School of Chemical Engineering, Engineering Campus, Universiti Sains Malaysia, Nibong Tebal 14300, Pulau Pinang, Malaysia; aabiodun581@gmail.com; 2Process Engineering Department, International Maritime College, Sohar 322, Oman; jimoh@imco.edu.om

**Keywords:** mixed matrix membranes (MMMs), carbon dioxide, lignocellulosic biomass, pretreatment, mechanism

## Abstract

In this paper, a review of the compatibility of polymeric membranes with lignocellulosic biomass is presented. The structure and composition of lignocellulosic biomass which could enhance membrane fabrications are considered. However, strong cell walls and interchain hindrances have limited the commercial-scale applications of raw lignocellulosic biomasses. These shortcomings can be surpassed to improve lignocellulosic biomass applications by using the proposed pretreatment methods, including physical and chemical methods, before incorporation into a single-polymer or copolymer matrix. It is imperative to understand the characteristics of lignocellulosic biomass and polymeric membranes, as well as to investigate membrane materials and how the separation performance of polymeric membranes containing lignocellulosic biomass can be influenced. Hence, lignocellulosic biomass and polymer modification and interfacial morphology improvement become necessary in producing mixed matrix membranes (MMMs). In general, the present study has shown that future membrane generations could attain high performance, e.g., CO_2_ separation using MMMs containing pretreated lignocellulosic biomasses with reachable hydroxyl group radicals.

## 1. Introduction

The world population as estimated in 2020 was about 7.7 billion, with projected increases of about 30% by 2050 and around 2100, to 9.7 billion and 11.2 billion, respectively [1]. Enormous challenges accompany this. Of our primary concerns are polluting gases from different processes and industries that are emitted into the atmosphere. The well-known group of gases, greenhouse gases, includes carbon dioxide, methane, ozone, carbon fluorocarbons, and nitrous oxides as the main ones. If present in the atmosphere, these gases trap and radiate heat and absorb infrared radiation. The effects of these actions are evidenced in the climate problem and retention of heat during nighttime. The comparisons made among these gases have led to the recognition of the adverse effects from the amount of released carbon dioxide and have made it the main acid gas to be investigated to reduce greenhouse gases’ effects [2,3,4].

According to the report from the Global Monitoring Laboratory of Earth System Research Laboratories at the National Oceanic and Atmospheric Administration, CO_2_ emission on a daily average basis for some locations is represented in Figure 1, which shows the average global levels. About 450 ppm concentration of CO_2_ is present in the atmosphere [5]. Consistent rise in the emission of CO_2_ from energy use, of about 10%, based on the different scenarios of transition evolving around us, will continue until 2040, based on the claims of the Energy Outlook from BP [6,7]. The introduction of carbon compounds in billions of tons into the atmosphere from the beginning of the industrial revolution until now will require a robust approach to mitigate. At present, CO_2_ elimination in the atmosphere is not exceeding half of the emitted amount, while the destruction of ozone layers and global temperature rise are among the harmful effects of leftover CO_2_. Prompt actions are required to solve this CO_2_ menace because it is affecting industrial activities such as reduction of flue gas heating values, industrial catalyst deterioration and poisoning, and transfer pipeline corrosion [8]. Therefore, a comprehensive and collective approach will be required for CO_2_ separation from the flue and natural gases with high purity and efficiency. Also, international laws and regulations must be enforced as mitigation measures while researchers are busy working on carbon capture, storage, and utilization.

An agreement was signed at the climate change convention held in Paris (2016) to increase the awareness of the negative impacts of CO_2_ on changes in the climate and to ensure that the temperature rise is limited to below 1.5 °C. The decision was made based on the outcomes from the earlier conventions on climate change and the Kyoto conventions held in 1992 and 1997 [9,10,11]. Natural gas purification by the removal of acid gases especially has passed through different separation processes such as absorption, adsorption, membranes, and cryogenic distillation. A description of each method can be found elsewhere [12,13]. Yang et al. [14] carried out the economic comparison for the power plant flue gas CO_2_ capture. They posited that the most cost-effective method is chemical adsorption and suggested that improving the performance of membranes can be an alternative for future promising gas separation. Preference given to membrane technologies in gas separation is revealed in Table 1.

## 2. Membrane Materials and Gas Separation Mechanism

The membranes for gas separation are two phases that are adjacent to each other acting as an interphase or selective barrier regulating the transport of gas mixtures. Membrane separation is an environmentally friendly process projected to be necessary as a solution to environmental problems, because the separation technique has an excellent downsizing footprint and easy operation. The gas separation process using membranes depends on the choice of materials to be used for membrane fabrications. The types of selected materials will affect the permeability, the structure/thickness of the membrane, and the choice of the design module. The module design configuration can be flat or hollow [31]. A significant gas separation using membrane technology favors the separation of gas molecules with different molecular sizes. Investigations have been intense to separate some perfluoro compounds (e.g., SF_6_) and the following combinations: CO_2_/CH_4_, CO_2_/N_2_, O_2_/N_2_, etc. In nanometers (nm), the kinetic diameters of the common gases H_2_, CO_2_, N_2_, and CH_4_ are 0.29, 0.33, 0.36, and 0.38, respectively. These values indicate the influence of the spherical size of the gas molecules, which is related to the mean free path of molecules in a gas. The mean free path of gas molecules connotes the average distance travelled by a particle without collision. Thus, a smaller kinetic diameter means a higher likelihood or possibility of collision between a gas molecule and another molecule [32,33]. The practical application of polymeric membranes has been established with other separation technologies such as dialysis and reverse osmosis. However, this field of membrane technology demands further development to achieve excellent separation performances [34].

Performance superiority of membranes is determined by the permeability level and selectivity for a targeted gaseous species during gas processing operations. Application is given significant consideration among other factors such as cost, feed solution composition, goals of separation, parameters of operation, and membrane fabrication technology compatibility before choosing between an inorganic or organic barrier. The polymeric membrane material development is inevitable, since the desired separation performance is yet to be achieved.

An important class of membrane materials that have received a lot of attention is the mixed matrix membrane (MMM) material. Composite (mixed matrix) membranes are obtained when a filler material (usually in solid phase) is integrated into a continuous matrix of the polymer [35]. The filler can be either an organic or inorganic material. The main aim is to capitalize on the micropores of inorganic fillers that may provide better interaction with any of the gas components, because not all inorganic fillers have good interaction with targeting gases [36]. Thus, by incorporating an organic or inorganic filler into a continuous polymer matrix, researchers can exploit the characteristic synergy that results from the nature of the microstructures of both materials.

The continuous polymer matrix and the organic fillers (which can be either synthetic or natural) are glassy and rubbery polymers of varying properties (Table 2) [37]. Glassy polymers are chemically and thermally stable. In turn, these properties reflect in their processability and permselectivity results. Polyethersulfone, polyetherimide, polycarbonate, poly(2,6-dimethyl-1,4-phenylene oxide), polymers of intrinsic microporosity (PIMs), polyimide, sulfonated poly(ether ketone), and cellulose acetate are some examples of glassy polymers [32]. The common rubbery polymers are polydimethylsiloxane and propylene oxide–amide copolymers. Therefore, careful selection of the polymer to be used demands thorough investigations.

Stereoisomerism, the polarity of functional group, rigidity, and interaction of chains must be considered to make the right choice of polymers for the manufacture of organic membranes. Two or more monomers can be polymerized to synthesize artificial polymers. The polymerization could be by using either of the three configurations: linear (e.g., polyethene), branching (e.g., polysulfone), or crosslinking (e.g., phenol–formaldehyde) chains/structures. Linear-chained polymers (also called thermoplastic polymers) are easy to mold at high temperatures and dissolve readily in organic solvents. However, the increasing temperature does not soften crosslinked polymers (also called thermosetting polymers) and gives slight dissolution in organic solvents. Ceramics and metals constitute the inorganic barriers. Ceramic membranes entail titanium or aluminium metals and carbides, nitrides, and oxides of nonmetals. These ceramic membranes possess inert properties that enhance their applications in environments that are overly acidic or basic. Cracking of these membranes due to their high-temperature sensitivity is a downside. As for metallic membrane formation, powdered stainless steel, palladium, or tungsten metals are deposited and sintered on a porous substrate. The best combinations of polymer, solvent, and nonsolvent are reported elsewhere for gas separations [23,38,39,40]. Figure 2 represents the dope formation and post-treatment flow chart to obtain defect-free membranes. In general, for organic membranes, it is very challenging to clean the fouling surface, as it is chemically and thermally unstable and the material is prone to degradation by microorganism. In contrast to organic membranes, inorganic membranes require differences in pressure drop to be accounted for by ensuring a specified thickness is fabricated, which usually results in a higher cost [41].

The separation of gases across membranes is driven by the pressure differences between the feed (inlet raw material) and the product (outlet). If the system is a gas–fluid phase with the gradient of partial pressure as the driving force, the membrane structure should be dense/porous for efficient gas permeation evaluation [45,46]. Permeability and selectivity influence membrane performance. The size of the penetrants will affect the diffusion coefficient and permeability because the gas molecules will have sufficient space to move, as the polymer has free volume and its chains are flexible. The membrane selectivity is a function of the ratio of the membrane’s permeability for relevant gases. Therefore, understanding the properties of the gas transport mechanism for a gas–fluid system is essential.

The mechanistic approach to transport (mass or energy) through a membrane depends on the module used. A module, also called the separation unit, is the specified smallest unit for packing membrane area. Installation of a membrane requires that the module is given priority, and using a single module is the simplest design. The permeate and retentate streams are separated feed streams that pass through as distance increases, which results in the decrease in both flow rate and concentration inside the module (Figure 3). Flat (e.g., spiral-wound and plate-and-frame modules) and tubular (e.g., hollow fiber, capillary, and tubular modules) configurations of membranes are used by researchers to design various modules [47]. The diameter of fiber size (in mm) influences the choice of a module; if the diameter is less than 0.5, between 0.5 and 10.0, or more than 10.0, the module choice will be hollow fiber, capillary, or tubular, respectively. A detailed description of these modules can be found elsewhere for spiral-wound [48,49,50], plate-and-frame [51,52,53], hollow fiber [24,54,55,56,57,58,59], capillary [60,61,62,63,64], and tubular [65,66,67,68,69,70,71,72,73] module configurations. The membrane module in the equipment receives the pumped gas, and the diffusivity and solubility differences enhance the targeted gas separation transport mechanism. In general, a module could be dead-end or crossflow (Figure 3). Microfiltration is frequently carried out using the dead-end module. In this, the membrane experiences feed forced on it, while the rejected component concentrations at the feed increase and permeate quality decreases with time. For instance, the downstream and upstream sides during ambient air gas separation using membrane produced nitrogen and oxygen, respectively [74]. The solubility coefficient ($S_{i}$) is estimated by dividing the gas concentration in the polymer with the gas partial pressure in contact with it (Equation (1)), and its contributions to membrane transport mechanisms cannot be ignored.
(1)$$S_{i}=C_{i}{\left(p_{i}\right)}^{-1}$$

The primary transport mechanisms are Knudsen diffusion and molecular sieving for pored membranes, and solution diffusion for dense membranes. Molecular sieving is designed to prevent specific larger molecules from passing through its small pores. Knudsen diffusion is applicable when pressure is low while the pore size of the material is relatively large and smaller molecules move faster through the pores than the larger ones. Both molecular sieving and Knudsen diffusion mechanisms are not practicable for gas separation because their molecular flow is convective and driven by pressure through a capillary and can be estimated using Darcy’s law. The solution-diffusion mechanism is the most appropriate for gas separation (Figure 4). The diffusion coefficient ($D_{i}$) is described using Fick’s law (Equation (2)). Therefore, the product of Equations (1) and (2) can be used to evaluate the permeability (Equation (3)), where $C_{i}$ can be the gas concentration for the feed or the permeate sides as $C_{i,\;i}$ and $C_{i,f}$, respectively; while $p_{i}$ can be the partial pressure for the feed or the permeate sides as $p_{i,i}$, and $p_{i,f}$, respectively.
(2)$$D_{i}=J_{i}l{\left(S_{i}△P\right)}^{-1}$$
where $J_{i}$ is flux from Fick’s law, l is the thickness of the membrane, and $△P\;\left(p_{i,i}-\;p_{i,f}\right)$ is the difference in partial pressures from the feed and permeate sides.
(3)$$P_{i}=D_{i}S_{i}$$

Permeability is mostly expressed in barrer ($1\;barrer=1\;\times 10^{-10}\frac{cm_{STP}^{3}\;cm}{cm^{2\;}\;s\;cmHg}$). In SI units, the expression for $1\;barrer=3.35\;\times \;10^{-16}\frac{mol\;m}{m^{2}\;s\;Pa}$. However, the penetrant gas molecular weight ($M_{r}$ in mol/g) must be accounted for while using the centimeter-gram-second (cgs) unit to estimate 1 barrer (Equation (4)).
(4)$$I\;barrer=M_{r}\;\times \;3.35\;\times \;10^{-13}\frac{cm\;g}{cm^{2\;}\;\;s\;\;\;bar}$$

Gas permeance ($P_{i}/l$), permeability divided by membrane thickness, is also commonly expressed in gas permeance units (GPU), where $1\;GPU=1\;\times \;10^{-6}\frac{cm_{STP}^{3}}{cm^{2\;}s\;cmHg}=1\;\times \;10^{-12}\frac{m_{STP}^{3}}{m^{2}\;s\;Pa}$ and $M_{r}$ (in mol/g) must be incorporated for the penetrant gas molecular weight for the SI unit conversion (Equation (5)).
(5)$$1\;GPU=M_{r}\;\times \;3.35\;\times \;10^{-10}\frac{mol}{m^{2}\;\;\;s\;\;Pa}$$

The membrane selectivity (second vital parameter) ($\alpha_{x/y}$) is the membrane’s separating ability, and it depends on the diffusion rate of the specified gas molecules. In this situation, it implies the ratio of the permeability of penetrants x and y (Equation (6)). $D_{x}{\left(D_{y}\right)}^{-1}\;$ and $S_{x}{\left(S_{y}\right)}^{-1}$ represent the diffusivity and solubility selectivities, respectively. The ratio of the size of the molecules influences the diffusivity selectivity (dominant) in glassy polymers. As for rubbery polymers, solubility selectivity gives the major contribution. However, for both glassy and rubbery polymers, when the fractional free volume is similar, they exhibit the same diffusivity and selectivity. Table 3 presents the permeability and selectivity trade-off for some polymers. In ideal situations, the upper bound lines are used for determining the best materials for membranes based on their separation performances [76,77,78].
(6)$$\alpha_{x/y}=P_{x}\;P_{y}{}^{-1}=D_{x}S_{x}\;{\left(D_{y}S_{y}\right)}^{-1}=D_{x}{\left(D_{y}\right)}^{-1}\;S_{x}{\left(S_{y}\right)}^{-1}$$

For a low-recovery system, assuming temperature and permeability coefficients are constant, the gas separation conditions will be similar to the counter-flow. The feed side experiences plug flow while the permeate side is completely mixing, and it implies that mixing is thorough at the feed and permeate sides. The area of the membrane can be estimated by using Equation (7).
(7)$$A=\frac{q_{p,i}}{J_{i}}=\frac{q_{p\;\;x_{p,i}\;}}{J_{i}}$$
while $J_{i}$ can be obtained from Equation (2), $q_{p}$, and $q_{p,i}$ are the permeate total mass flow rate and the mass flow rate of component *i* in the permeate, respectively.

It is worth mentioning that a membrane unit’s capital cost can be reduced by reducing membrane thickness or modifying the chemistry of the membrane, which will result in membrane system size reduction and higher permeance. More so, the membrane system operating cost will be reduced since it is related to the required energy and the energy consumption depends on membrane selectivity. The majority of polymers used for membrane fabrications contain both crystalline and amorphous fractions. However, higher crystallinity results in higher resistance to diffusion. Reducing the amount of crystallinity in the membrane will affect the flux, diffusion rate, and transport mechanism of the membrane (Equation (8)).
(8)$$D_{i\;}=\;D_{i,0\;}\;\;B^{-1}\;\;\psi_{c}^{n}\;\;$$
where $\psi_{c\;}$(<0.1) is the crystalline amount present and n (<1) and B are an exponential factor and constant, respectively. So, gas performance improvement could be improved by using some techniques to pretreat the polymer membranes. These techniques are polymer backbone grafting [86,87], porogens usage with polymerization template [88,89], thermal and crosslinking [90,91], blending and copolymerization [92,93], phase inversion [94,95], polymer sulfonation [96,97,98], and the use of PIMs [99,100].

As a form of polymer pretreatment, Kang et al. [101] suggested surface binding increment for CO_2_ on polymers to improve gas separation by tuning metal ions. Also, Fontaine et al. [102] posited that CO_2_ with its Lewis acidic properties facilitates separation affinity, and Kundu et al. [103] stated that the uptake of CO_2_ would be massive in the presence of polymers with hetero elements that are rich in electrons, such as –C=O. Figure 5 shows the interaction of CO_2_ with polymers of pyridine and imidazole. The chemistry is due to the lone-pair electron nucleophilic attack. Therefore, this is a justification for the surface modifications to improve performances (permselectivity) by the incorporation of organic/inorganic materials (fillers) such as lignocellulosic biomasses into polymers to enhance their reaction mechanisms.

## 3. Lignocellulosic Biomasses Retrospect

An annual estimation of total global agricultural waste materials produced in 2017 was about 37,522,440,479 kg, with lignocellulosic biomass topping the list, which implies that large quantities of agro-wastes are available to be turned into resources [105]. The efficient use of lignocellulosic biomass requires significant study, understanding, and separation of its major complex components. Weak Van der Waals and hydrogen bonds hold these three components (lignin, hemicellulose, and cellulose) together within the plant cell walls [106].

The middle lamellae of opposite cell walls contain lignin, and it serves as the hydrophobic surface for water transport in plants to about 100 m height. Lignin also gives trees exceptional mechanical support of almost two kilometric tons and prevents invasion of pests and pathogens due to its chemical and physical properties [107,108,109,110,111]. However, high-value utilization of biomass has been faced with challenges attributed to rigidity, which leads to deconstruction resistance and recalcitrance affecting technological and economic conversion applications. Lignin poses a physical obstacle to direct access to cellulose, and biological and enzyme activities on biomass are diminished due to this recalcitrance [112,113,114,115]. Biofuels, paper, and pulp productions will increase if lignin presence in biomass can be effectively reduced. Nonproductive enzyme-to-lignin binding is another undesirable mechanism showing adverse effects of lignin on catalyst applications. This interaction between lignin and enzymes can be quantified by using the Langmuir adsorption isotherm analysis [116,117].

Different parts of trees contain varying lignin content with variable compositions: the shoots and wood have low and high lignin content, respectively [118,119]. There are three lignin subunits (Figure 6): First, guaiacyl units (G) form softwood lignin and its polymer is derived from the monomers of coniferyl alcohol. Second, syringyl units (S) are formed from the polymerization of the monomer of sinapyl alcohol. Third, *p*-hydroxyphenol (H) phenylpropanoid is a mixed S and G unit, which is common in hardwoods depending on the S/G ratio [119]. Lignin also contains aromatic units (inset of Figure 6), namely coumaryl (a), coniferyl (b), and sinapyl (c) alcohols. The plant species have different units, and the bonding also differs. All plant species, especially softwoods, contain coniferyl alcohol. Crops and grasses mainly consist of coumaryl alcohol. About 40% of alcohol units in hardwoods are sinapyl units. Powerful C–C bonds and ether linkages keep the lignin structure together, which makes lignin insoluble in water. The reaction mechanism of lignin is challenging because its structure is highly complicated. Researchers are working on different routes to lignin breakdown. Some of these routes are ionic liquid usage, lignin esterification, supercritical carbon dioxide, and pyrolysis [120,121,122,123]. Hemicellulose and cellulose are less energy-intensive compared to lignin, as we will discuss next. Kumar et al. [124] established that lignin removal increases hemicellulose and cellulose accessibility, which will enhance their OH groups’ availability for different applications. Wang Y. et al. [125] and Wang S. et al. [126] studied the effects of pretreatments on breaking lignin structural linkages to increase accessibility by increasing internal surface area.

Hemicelluloses are the second most abundant renewable polymers in lignocellulosic biomass after celluloses. Hemicelluloses are heterogeneous polysaccharides found in cell walls of plants. They have short chains, and are found in varying substituents and proportions [128,129,130]. Various applications can benefit from hemicellulose because they can be easily hydrolyzed, converted, and transformed [131]. Nonetheless, hemicelluloses also contribute to the resistances to the valorization of biomass. Cellulose extraction and quality of fiber and wood are affected due to hemicellulose presence. From the reaction mechanism, hardwood has different proportions of hemicelluloses with standard O-acetyl-4-O-methylglucuronoxylan components. At the xylopyranose backbone’s second and third carbon, the hydroxyl group is about 70% [132,133]. O-acetylgalactoglucomannan is the main component of hemicellulose in softwood. It consists of a 20% acetylated hydroxyl group, which was substituted partially by acetyl groups at the mannose units C-2 and C-3 [134,135]. Figure 7 shows hemicellulose sugar monomers that are of lower quantity in wood tissues and higher in some fruits’ soft tissues and the pulp of sugar beet.

Cellulose is one of the main plant cell wall components, rendering the mechanical strength, and is the most abundant due to the continuous photosynthesis process [137,138]. Several sources of available routes to cellulose are known as agricultural waste and are yet to be fully maximized for other applications. However, the manufacturing of textiles, cardboard, paper, pharmaceuticals, biofuels, food, and nanocellulose materials have benefitted from the exciting chemistry of cellulose to some extent [139,140,141,142]. Cellulose’s chemistry and mechanism depend on its high polydispersity. It is closely connected to other biopolymers such as lignin and hemicellulose. These other biopolymers have a challenging spatial arrangement. One glucose monomer in cellulose is called anhydroglucose units. Cellulose is a linear isotactic polymer with 180° rotated neighboring monomer attachments. The attachment is made up of two glucose units (called cellobiose, repeating units). The monomer of glucose in cellulose is described as being in a chair conformation. The nonreducing end contains a glycosidic bond within its carbon atoms, while the reducing end converts its carbon to aldehyde. These two ends give balance (equilibrium) to cellulose [143,144,145,146]. The crystallinity and hydrogen bonding of cellulose is impacted. This impact is attributed to the numerous hydroxyl (OH) groups present in its metastructure (Figure 8).

Therefore, pretreatment is vital for the production of value-added products from lignocelluloses. It means that fractions of cellulose and hemicellulose should be made more accessible. The cellulose digestibility should also be increased by following sets of steps. The targeted actions aim at fractions of lignin to be bond-broken, solubilized and removed (Figure 9) [148,149,150]. Physical, chemical, physicochemical, biological methods and their combinations are the general pretreatment classification processes [151]. Enzyme, feedstock, and organism compatibility influence the pretreatment choice. The detailed lignocellulosic biomass pretreatments are beyond the scope of this review and can be found elsewhere [152,153,154,155]. However, an overview is given in Table 4.

Finally, complex components of lignocellulosic biomasses can be solubilized by combining these pretreatment methods. The outputs from physical and chemical pretreatments were relatively good even though they were attributed to severe pollution and strict equipment requirements. Although they release less pollution compared to other methods and have lower energy consumption, biological processes are time-consuming and expensive. Li et al. [168] studied different techniques to alter the lignin contents in crops to improve cellulose and hemicellulose accessibility. The primary outcome of their study was that genetic engineering of lignin biosynthesis relied on monolignol pathway up- or down-regulation. Plant fitness, viability, and biofuel applications will be disfavored when total lignin is reduced. Finally, plant manipulation to ease enzyme or alkaline hydrolysis of lignin linkages will improve biomass applications.

## 4. Perspective and Prospects for CO_2_ Separation

Lignocellulosic biomass pretreatment becomes necessary. The necessity is due to an increase in the accessible surface area and the hydroxyl groups, lignin removal, and cellulose decrystallization. One of the ways to solve the trade-off of polymeric membranes is the incorporation of organic/inorganic materials to produce mixed matrix or composite membranes [169,170]. The lignocellulosic biomass-based MMM hybrid system begins with the targeted gas molecules being sorbed by the membrane. The adsorption occurs at high pressure due to the interaction with the filler (lignocellulosic biomass), diffuses through the membrane and then desorbs at the low pressure. The thermodynamic parameter (solubility coefficient) and the kinetic parameter (diffusivity coefficient) control this mechanism. As discussed in earlier sections, gas separations and other applications have benefitted from these concepts. This review examines the compatibility of polymers and lignocellulosic biomass. Cellulose extraction has been performed by solubilizing lignin, followed by incorporation into a polymer matrix after improving its intermolecular and intramolecular bonding [171]. These easily accessed OH groups will aid carbon dioxide capture because of the mechanism between CO_2_ and OH radicals. Table 5 presents the extraction techniques of cellulose.

Mixed matrix membranes for various applications have been fabricated by incorporating cellulose extracted from natural agro-based sources into polymers, carbon, etc. (Table 6). A search of electronically available literature with keywords such as lignocellulose and membranes revealed that most biorefining research works are focused on biofuel, bioethanol, etc. The forerunner in the valorization of lignocellulosic biomass is the ChemPubSoc Europe organization. This organization constitutes 16 chemical societies. It shows the importance of lignocellulosic biomass as an alternative feedstock for energy, environmental, and crude oil substitute applications. The valorization will open the doors for interdisciplinary collaborations with colleagues working on catalysis and reaction engineering, among others. So, its usage as fillers in membrane fabrications cannot be overemphasized.

The polymer forms the continuous phase, while the cellulose forms the dispersed phase. Therefore, the polymer and filler must be carefully selected, since a defective morphology is obtained when the rigidification of the surrounding polymer matrix occurs due to the effect of the dispersed phase causing a void at the continuous and dispersed interface. Gas separation, water purification, tissue engineering, food packaging, etc. enjoy the remarkable properties of the mixed matrix membranes from cellulose due to their excellent mechanical strength. Elrasheedy et al. [186] reviewed water applications of membranes. Also, O’Harra et al. [187] fabricated composite membranes for gas separation. Supercapacitor electrodes are produced from carbon nanotube and cellulose combinations. Researchers are working on modifications of polymers and extracted cellulose for environmental, gas separation benefits, etc., including modifying cellulose to reduce its crystallinity and particle size to the nanoscale, increase its tensile strength and access to its hydroxyl group radicals, and use several functional groups. Polymers have been modified using different functional groups, such as amine [188], aniline [189,190], methacrylate [191,192,193], polyvinyl alcohol [194,195], and polyethylene oxide [196].

Table 7 presents some filler effects on the CO_2_ separation performance in MMMs, and it establishes that the filler compatibility with the polymer matrix was owing to their small particle sizes. Also, the transport of CO_2_ was facilitated because the polymer chains’ free volume and packing were influenced by the fillers.

Table 8 shows the performance results of the fabricated mixed matrix membranes. Pretreated lignocellulosic biomass-based filler was loaded into the polyetherimide (PEI) matrix. A combination of physical and chemical pretreatment techniques was used to obtain a functionalized filler. The powder X-ray diffraction showed a decrease in its crystallinity. In almost all cases, CO_2_ permeability increase was more evident than that of CH_4_. The Langmuir model-governed transport mechanism ensures the saturation of the free volume due to the CO_2_ and CH_4_ gases’ kinetic diameters.

In contrast, the transport mechanism will be governed by Henry’s mode whenever CH_4_ permeability is favored over CO_2_, due to gas–polymer interaction increase and larger molecules being easily condensable. The pure PEI membrane shows low permeance and selectivity compared to after the filler was incorporated. However, 0.5 wt% filler loading results in the best permeability–selectivity performance. n The permeability–selectivity decrease recorded for 1 wt% loading can be attributed to less filler dispersion, and this can be resolved during doping preparation by including the sonication step because that effect is attributed to the dispersed phase clogging of the interface of the two phases. Considerations would be given to the shape and size of the particles and the preparation protocols.

Also, Wu et al. [225] worked on CO_2_ separation by incorporating biocellulose nanofiber (additive) into polymeric membranes. A homogeneous composite membrane solution was obtained by speed coating the additive using the spin coating method and oven drying of the membranes for half an hour at 105 °C. The results showed that increasing polymeric solution concentration resulted in increasing the thickness of the membranes. The performance (permeability and selectivity) for 3 wt% loadings of the additive was chosen as the best. Zhang et al. [226] reported 139 barrer CO_2_ permeability from composite membranes with incorporated cellulose nanofibrils, adopting chemical pretreatments to enhance the chemical bonding for high performance.

As shown in Table 9 [36], it was posited that at 5 wt%, permeability was higher than at 2 wt% loadings on polysulfone membranes because it was more amorphous, which was further confirmed by the XRD results. Although, some of the selectivity results are not in line with established trends from the Robeson upper bound curve: to have selectivity increasing as permeability decreases. The upper bound trend shows a gradual decline in permselectivity. The addition of 2 wt% Lignocellulosic biomass (LCB) to the polymer matrix increases the selectivity, which then started decreasing as loading increased from 5 to 10 wt%. In general, the simultaneous decrease in permeability and selectivity can be attributed to polar and nonpolar gases being separated based on the competitive sorption that favors CO_2_ due to its polarity, which is one of the expected setbacks from the separation of gases with close molecular diameters. CO_2_ is a polar gas molecule, while CH_4_ is a nonpolar gas molecule. Lastly, Venturi et al. [227] used a combination of mechanical and enzymatic pretreatment techniques to prepare nanofibrillated cellulose and incorporate it into polyinylamine (PVAm) to fabricate composite membranes. Values of 135 and 187 barrer were reported for CO_2_/CH_4_ ideal selectivity and CO_2_ permeability, respectively.

Finally, since the lignocellulose biomass contents are natural, degradation resistance and stability can be made robust through the pretreatment techniques, which will also prevent fouling. Thus, a thorough understanding of the lignification mechanism before incorporation into the polymer matrix is vital. Prospectively, MMM fabrication by incorporating lignocellulosic biomass will be faced with some challenges. The challenges include discovering an environmentally and cost-effective benign delignification process. Furthermore, process condition optimization and improvement of existing methods for the different pretreatments’ reaction mechanisms have not been thoroughly explored. Multidisciplinary collaboration among material chemists, polymer scientists, and chemical and material engineers is required. Such partnership will enhance worldwide acceptance of the development of next-generation membrane materials, looking back to 1979 when Monsanto built the first hollow-fiber polysulfone membrane separation system. The next-generation materials should be able to withstand physical ageing, plasticization, etc. to increase their consideration for industrial applications. Therefore, this review discusses the characteristics of membranes and lignocellulose biomass, including the feasibility of solubilizing lignin. Some of the common challenges include extracting or isolating cellulose and hemicellulose from several agro-based sources such as wood, straws, grasses, etc. Thus, it is imperative to carefully choose an appropriate pretreatment technique(s) for high purity, yield, and compatibility; and reducing the particle size to the nanoscale using mechanical splintering will aid the reduction of the crystallinity zone for subsequent treatments. The OH group’s characteristics are fundamental to lignocellulosic biomass performance and compatibility during incorporation into the polymer matrix.

## Figures and Tables

**Figure 1 membranes-10-00370-f001:**
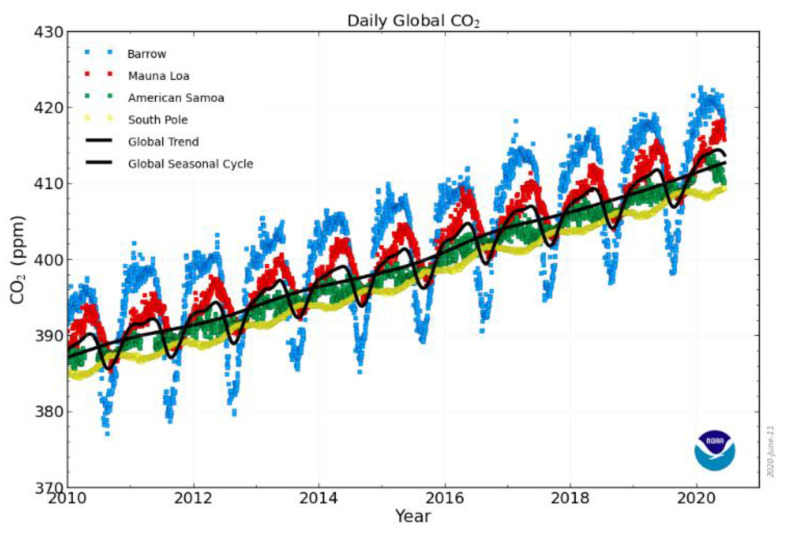
Recent trends in CO_2_ globally; reproduced from Reference [5].

**Figure 2 membranes-10-00370-f002:**
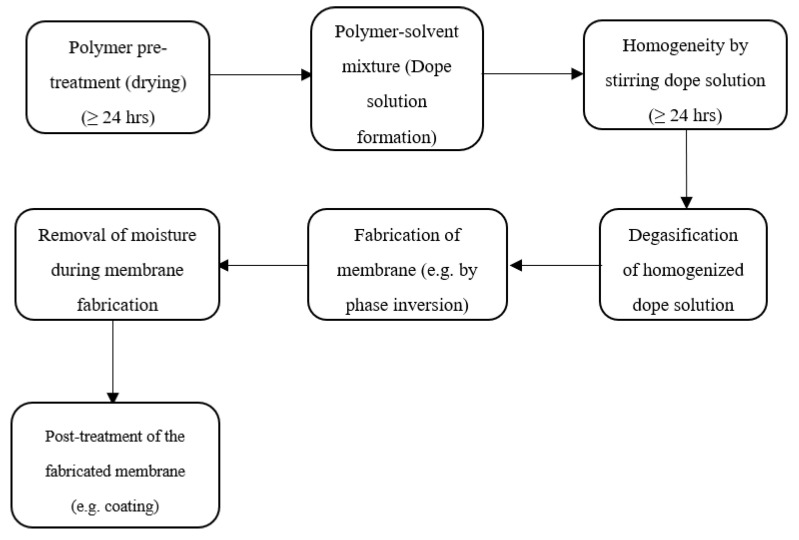
Defect-free membrane fabrication flow chart [41].

**Figure 3 membranes-10-00370-f003:**
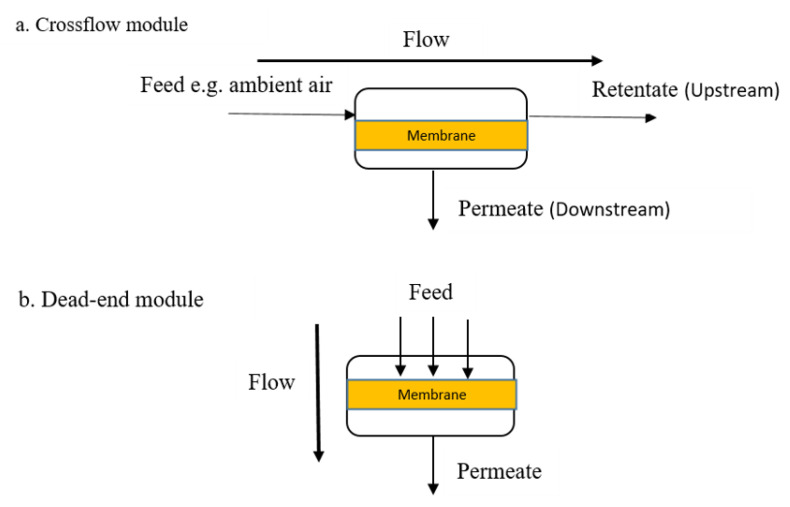
Basic schematics of two module operations: (**a**) crossflow module; (**b**) dead-end module.

**Figure 4 membranes-10-00370-f004:**
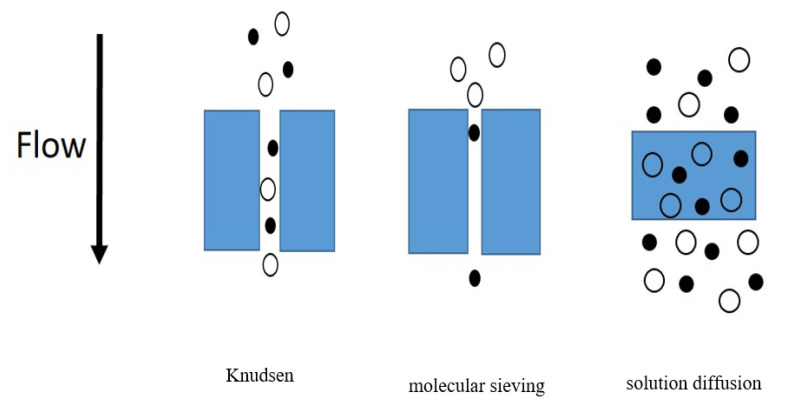
Schematic of membrane diffusion mechanisms; adapted from Reference [75].

**Figure 5 membranes-10-00370-f005:**
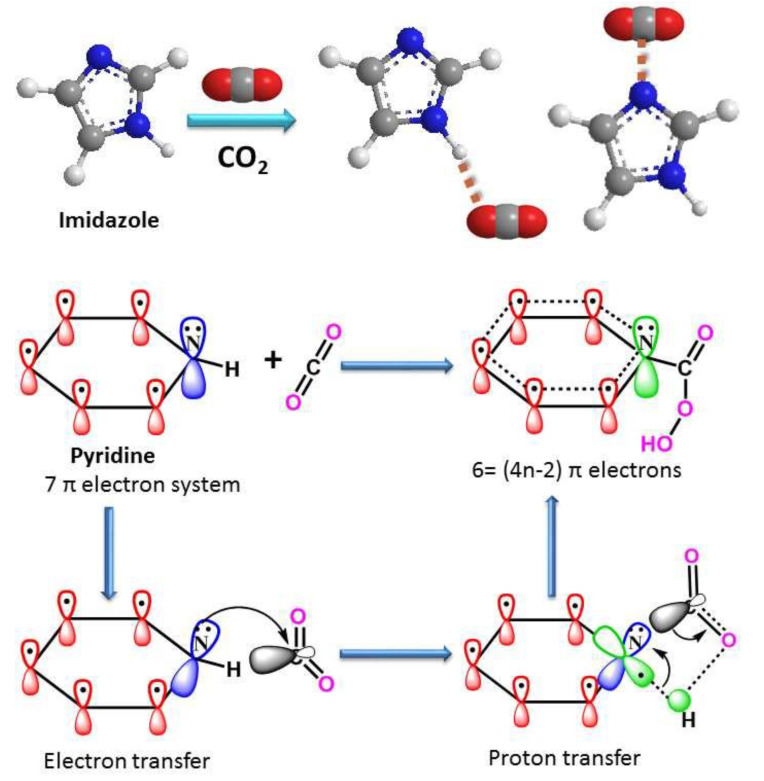
CO_2_ interaction mechanism with molecules of pyridine and imidazole polymers; reproduced from Reference [104].

**Figure 6 membranes-10-00370-f006:**
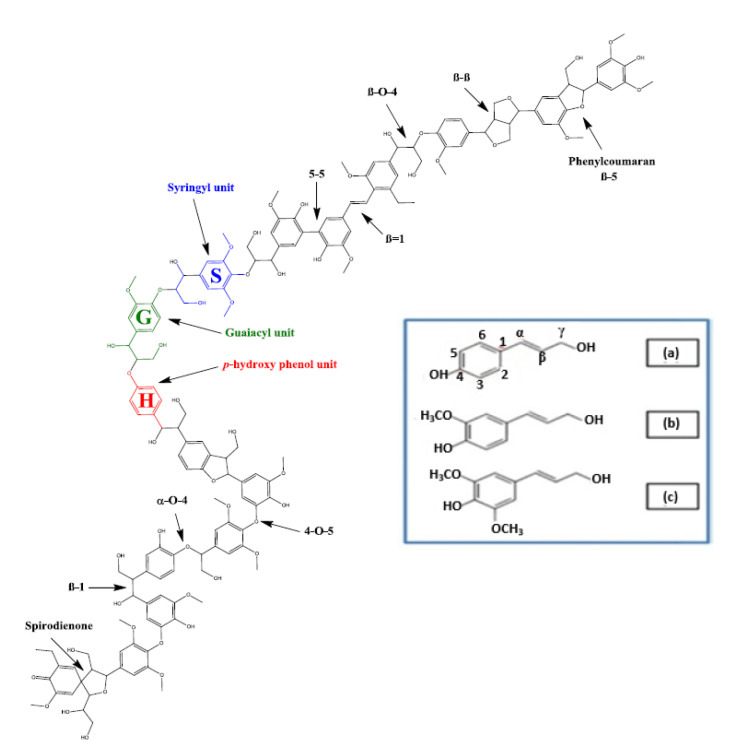
Subunits in the lignin structure; adapted from Reference [127].

**Figure 7 membranes-10-00370-f007:**
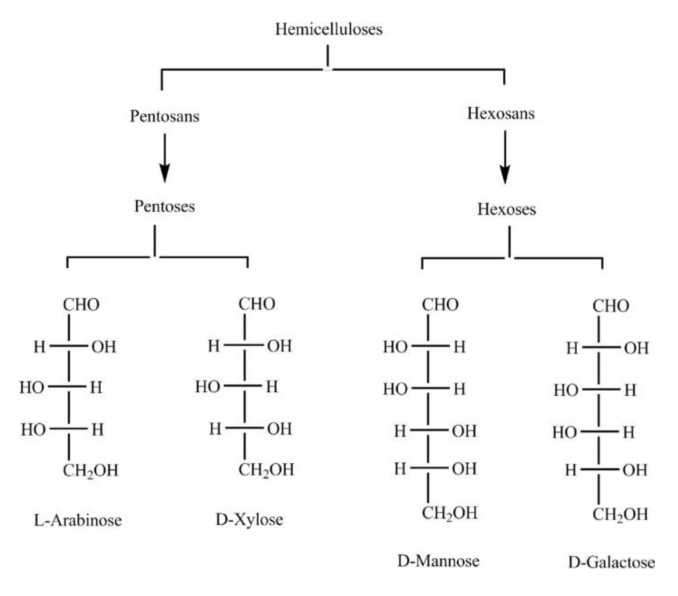
Hemicelluloses monomers; reproduced from Reference [136].

**Figure 8 membranes-10-00370-f008:**
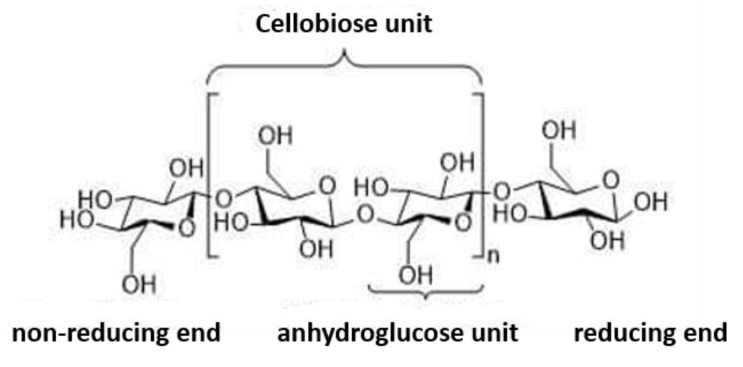
OH groups in the cellulose structure; reproduced from Reference [147].

**Figure 9 membranes-10-00370-f009:**
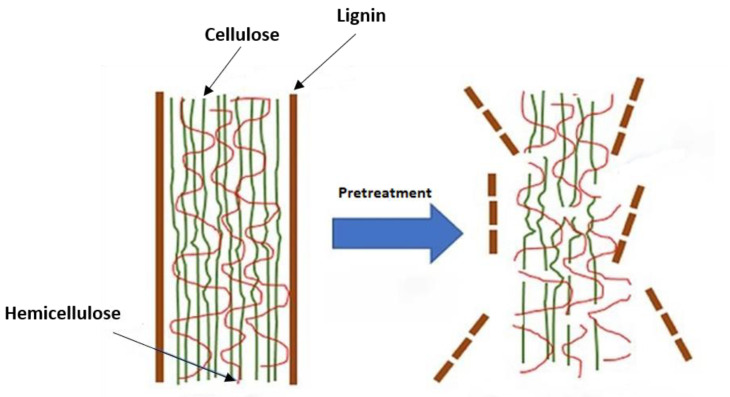
The role of pretreatment; reproduced from Reference [156].

**Table 1 membranes-10-00370-t001:** Comparisons of Natural gas (NG) separation processes.

Separation Process	Merits	Demerits	Ref.
Absorption	1. Acid gases (CO_2_ and H_2_S) removed simultaneously 2. Processing capacity and product purity is high 3. Efficiency is 50–100%	1. Physical solvent usage requires high partial pressure. As such, it is not cheap 2. Chemical solvent usage requires low partial pressure, which makes acid gas purification take a longer time 3. Operating units’ efficiency is low, especially using the amine absorption process	[15,16,17,18]
Adsorption	1. Can produce an output of high purity 2. Remote field relocation of adsorbent, at the time of equipment sizing challenge, is easy 3. Simple process	1. Lower product recovery despite the high amount of used adsorbent 2. Single pure product is most favored 3. Poor performance at low pressure 4. Adsorbent regeneration is expensive	[19,20,21,22]
Membrane	1. Simplicity, versatility, low capital investment, and easy operation 2. High-pressure stability 3. Product recovery is high 4. Optimized weight and space 5. Environmentally friendly	1. Permeate can be recompressed 2. Permselectivity trade-off 3. Purity is moderate as its capacity is low 4. Available membranes are thermally unstable	[23,24,25,26]
Cryogenic	1. Compared to other techniques, recovery is relatively higher 2. Product purity is also relatively high 3. Operation is possible at high volume and high pressure	1. Regeneration requires energy of high intensity 2. Scale-down is not economical 3. Being a closed system that is highly integrated, different feed streams operation is challenging	[27,28,29,30]

**Table 2 membranes-10-00370-t002:** Examples of organic and inorganic membrane classifications [40,42,43,44].

Organic Membranes		Inorganic Membranes	
Synthetic	Natural	Ceramic	Metal
Polytetrafluoroethylene	Rubber	Silica membrane	Palladium membrane
Polyvinylidene difluoride	Cellulose	Silicon membrane	Tungsten membrane
Polyamide–imide	Wool	Zeolite	Palladium alloy membrane
Polysulfone, etc.	Leather	Carbon	Nonpalladium membranes

**Table 3 membranes-10-00370-t003:** CO_2_/CH_4_ diffusion selectivities for some polymers.

Polymers	$\mathrm{Permeability}\;P_{CO_{2}}$ (Barrer)	$\mathrm{Diffusion}\;\mathrm{Coefficient}\;D_{CO_{2}}$ (10^−8^ cm^2^ s^−1^)	$\mathrm{Total}\;\mathrm{Selectivity}\;\alpha_{CO_{2}/CH_{4}}$	$\mathrm{Diffusion}\;\mathrm{Selectivity}\;D_{CO_{2}}$ $/D_{CH_{2}}$	Ref.
Kapton polyimide	2.7	0.56	46.0	11.9	[79]
Polysulfone	5.6	2.00	22.0	5.9	[79,80]
Polycarbonate	6.8	3.20	19.0	4.7	[79]
Polystyrene	12.4	8.50	15.8	5.5	[79]
Poly(ethylene terephthalate)	17.2	4.46	27.3	7.8	[79]
Polyisoprene	153.0	125.00	5.1	1.4	[79]
Silicone rubber	3800.0	2200.00	3.2	1.1	[79]
Sulfonated poly(aryl ether ketone) (SPEEK)	15.0	4.89	26.5	2.8	[81]
Thermally rearranged polymers	186.6	15.40	27.8	4.0	[82]
Polymers of intrinsic microporosity (PIMs)	3672.0	172.00	10.6	3.9	[82]
Cellulose acetate	218.0	-	13.8	-	[83]
Perfluoro polymers (Teflon AF 2400)	2200.0	-	5.6	-	[84]
Poly(ether-block-amide) (Pebax^®^)	187.5	-	7.3	-	[85]

**Table 4 membranes-10-00370-t004:** Comparison of methods of lignocellulosic biomass pretreatments.

Pretreatment Type	Examples	Uniqueness	Ref.
Physical	Mechanical, ultrasonic, high-temperature/energy pyrolysis, electron radiation, and microwave	It requires high energy and cost to reduce crystallinity and particle size	[157,158,159,160]
Chemical	Pretreatments using alkali, organosolv, acids (dilute and concentrated), oxidation, and ionic liquids	The extraction of pure components is at a high cost	[161,162,163]
Physicochemical	CO_2_ explosion by Ammonia fiber explosion (AFEX) method, steam explosion, and electrical catalysis	Occurrences at high temperature and pressure: hemicellulose solubilization, lignin transformation, and cellulose surface area increases.	[164,165]
Biological	Enzymolysis	Despite low hydrolysis rate and energy consumption, degradation of lignin and hemicellulose is achievable	[166,167]

AFEX: Ammonia fiber explosion.

**Table 5 membranes-10-00370-t005:** Cellulose sources and extraction techniques [172].

Cellulose Sources	Preparation Techniques	Particle Size	Remarks	Reference
Pineapple peel juice	Spray coating	NA	Enhanced spread factor	[173]
Trunks and fronds of oil palm, okra	Alkaline, electrospinning	Less than 500 nm	Binding and antioxidant activities increased	[174]
Canola straw	nanowelding	53 ± 16 nm	High transparency and biodegradability	[175]
Rice husk	Hydrothermal approach, acid–alkali treatment, mechanical disruption	30–40 nm	Thermally stable	[176]
Corn	TEMPO-mediated oxidation	NA	High strength, elastic modulus, and value of water retention	[177]
Achira	Acid hydrolysis, high-pressure homogenization	13.8–37.2 nm	Mechanically stable, biodegradable, and highly crystalline	[178]
Paper waste residue	Etherification of pulp, mechanical disintegration	10–100 nm	Thermally stable with high fibrillation potential	[179]
Banana peel	Chemical treatment, high-intensity ultrasonication	NA	Highly thermally stable with high crystallinity	[180,181]
Poplar wood powder, culms of Moso bamboo, rice straw, corn straw	Chemical pretreatment, high-intensity ultrasonication, high-pressure homogenization	5–20 nm	Highly crystalline and thermally stable	[182,183]
Seagrass species balls and leaves	Chemical treatment, fibrillation	5–21 nm and 2–15 nm	Transparent and biodegradable	[184]
Tomato peels	Acidified sodium chlorite, chlorine-free alkaline peroxide	260 ± 79 nm,	Highly crystalline	[185]

TEMPO: catalyst used in organic synthesis as an oxidant.

**Table 6 membranes-10-00370-t006:** Cellulose-based mixed matrix and other details.

Mixed Matrix Composition	Cellulose Source	Preparation Method	Particle Size	Remarks	Ref.
Natural cellulose, high-density polyethene	Needle leaf bleached kraft pulp (NBKP)	Mechanical disintegration, injection molding	NA	Mechanical strength increased	[197]
Natural cellulose, polylactic acid	Wheat straw	Chemo–mechanical treatment, high-speed homogenization	NA	High viscosity with increased crystallinity	[198]
Cellulose, starch	Rice straw	A chemo–mechanical method, film casting, salt leaching, freeze-drying	49–90 nm	Highly biodegradable	[199]
Polyurethane, cellulose	Rachis of the date palm tree (*Phoenix dactylifera*)	Mechanical treatment, high-intensity homogenizing, solvent exchange method	29 ± 9 nm	Tensile strength, thermal stability, high crystallinity	[200]
Cellulose, hemicellulose	Spruce sulfite pulp (commercially obtained)	Enzyme treatment, mechanical disintegration, filtration, drying	190 nm	Moderate tensile strength with high thermal stability	[201]
Cellulose, starch	Kenafbast fibers (*Hibiscus cannabinus*)	Solution casting	NA	Biodegradable and moderate elasticity	[202]
Cellulose, polyester resin	Softwood (*Pinus* sp.) and hardwood (*Eucalyptus* sp.)	Mechanical treatment	70–90 nm	Moderately crystalline and highly thermally stable	[198]
Unsaturated polyester, cellulose	Never-dried wood pulp (Nordic Paper, Sweden)	Mechanical treatment, template-based processing approach	100–200 μm	Highly sensitive to moisture and thermally stable with a high glass transition temperature	[199]
Cellulose, amylopectin	Spruce sulfite pulp (Nordic pulp and Paper, Sweden)	Enzyme degradation, mechanical treatment, disintegration using a microfluidizer	68 nm, 361 nm, 186 nm	Yield strength increased with moderate Young’s modulus	[200]
Cellulose, polyacrylamide	Fibrous cellulose powder CF11 (commercially obtained)	Acid hydrolysis	NA	Hydrophilicity and high mechanical strength with favorable thermal stability	[203]
Cellulose, polyaniline, carbon nanotubes	Bamboo powders from Moso bamboo	Chemical treatment, in situ chemical polymerization	10–30 nm	Foldable and flexible	[204]
Cellulose, multiwalled carbon nanotubes, polyaniline	Bamboo powder from Moso bamboo	Chemical treatment, solvent extraction, in situ polymerization	10–30 nm	High porosity and redox reversibility	[205]
Cellulose, carbon nanotubes, TiO_2_ nanotubes	Bamboo cellulose tissues	Mechanical treatment	10–30 nm	Increased mechanical strength and porosity	[206]
Cellulose, cadmium sulphate (CdS)	Natural cotton	Electrospinning, chemical bath deposition	100 nm	Photocatalytic activity is high with characteristic amorphous properties	[207]
Titanium dioxide (TiO_2_), cellulose, gold (Au), silver (Ag)	*Eucalyptus* pulp (USDA Forest Service, Forest Products Laboratory, Madison, WI, USA)	TEMPO-mediated oxidation, mechanical treatment	4–20 nm	Reusable, with improved photocatalytic activities and high tensile strength	[208]
Cellulose, quaternary ammonium	Softwood kraft pulp	Mechanical treatment	10–40 nm	Reusable and highly porous	[209]
Polyethylene-*b*-poly(ethylene glycol), cellulose	Cellulose nanofibers (Commercially obtained)	Spray drying, surface adsorption, extrusion	NA	High modulus tension	[210]
Cellulose, polyvinyl alcohol	Microcrystalline cellulose (commercially obtained)	Acid hydrolysis	10–65 nm	Thermally stable and water-resistant	[211]
Cellulose, polylactic acid	Cellulose nanofibers (Commercially obtained)	Solvent casting	28 ± 10 nm	High elastic and tensile strength and thermally stable	[212]
Cellulose, poly(lactic acid)	Nano Nevin polymer co. (Iran)	Solution casting method	21 nm	High degradation temperature, thermal stability, and crystallinity	[213]
Cellulose, starch, polyvinyl alcohol	Microcrystalline cellulose (commercially obtained)	Acid treatment, solution casting	20–35 nm	Excellent mechanical strength with high stiffness	[214]
Polyethene oxide, cellulose nanocrystal	Microcrystalline cellulose (commercially obtained)	Acid hydrolysis, high-pressure homogenization, electrospinning	149 ± 49 nm	High glass transition temperature and elongation at break	[215]
Cellulose, copper (Cu^2+^)	Cellulose sludge (commercially obtained)	Mechanical treatment, TEMPO-mediated oxidation	15–40 nm	Enhanced adsorption capacity of Cu^2+^, wettability, and hydrophilicity	[216]

**Table 7 membranes-10-00370-t007:** MMM performances for CO_2_ separation based on filler effects [32].

Organic Filler	Particle Size (nm)	Loading (wt%)	Polymer	Feed Gas	Operation Conditions	CO_2_ Permeability ×10^14^/mol·m·m^−2^·s^−1^·Pa^−1^	CO_2_/CH_4_ Selectivity	Ref.
Polyaniline nanosheet	Thickness: 40–60	17	Poly(vinylamine)	Pure gas	25 °C, 0.11 MPa, in humidified state	40.20 × $l^{b}$	12–20	[217]
Polyaniline nanorod	Diameter: 50 Length: 160	17	Poly(vinylamine)	Pure gas	25 °C, 0.11 MPa, in humidified state	53.67 ^b^	18–25	[218]
Nanohydrogels	~250	5,10,15,20	Matrimid^®^	Pure gas	30 °C, 0.2 MPa, in humidified state	4.56–9.31	52–61	[219]
Carboxylic acid nanogels	400	5,10,15,20,30	Pebax^®^	Pure gas	25 °C, 0.2 MPa, in humidified state	29.82–67.87	19–33	[220]
PEGSS ^a^	350–420	20	Matrimid^®^	Pure gas	30 °C, 0.1 MPa	0.28	50.29	[221]
Hypercrosslinked polystyrene	55	16.67	PIM-1	Pure gas	25 °C, 0.2 MPa	334.06	20.27	[222]
Microfibrillated cellulose	Diameter: 5–15	0–4	Polyvinylamine	Mixed gas	35 °C, 8 bar	13.00	410	[223]
Nanocellulose	Length: 130 ± 67 Width: 15.9 ± 1.8	0.5,1,1.5,2	PSF	Mixed gas	25 °C, 8 bar	45.07	29	[223]

^a^ PEGSS: poly(ethylene glycol)-containing polymeric submicrospheres. ^b^ Permeability is calculated by permeance multiplied by the separation layer thickness, *l*: the separation layer thickness, μm.

**Table 8 membranes-10-00370-t008:** Biomass-based filler loading effects on membrane performance.

Filler Loading (wt%)	Permeance (GPU)		Selectivity
	CO_2_ Permeability	CH_4_ Permeability	
0.00	1.25	0.10	23
0.25	1.60	0.10	32
0.50	1.80	0.05	43
0.75	-	-	-
1.00	1.75	0.05	40

GPU: gas permeance units. Source: Reference [224].

**Table 9 membranes-10-00370-t009:** Permselectivity of pretreated Lignocellulosic biomass (LCB) (date pits) loading on CO_2_ separation.

LCB in Samples (wt%)	Permeance (GPU) at 10 bar and 35 °C		Selectivity
	CO_2_ Permeability	CH_4_ Permeability	
0	240.292	318.229	0.755
2	13.957	11.957	1.261
5	94.949	115.697	0.821
10	445.658	705.246	0.632

LCB: Lignocellulosic biomass.

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
