# Peer review of "Mechanism and Compatibility of Pretreated Lignocellulosic Biomass and Polymeric Mixed Matrix Membranes: A Review"

_membranes, 2020, doi:10.3390/membranes10120370_

Round 1
Reviewer 1 Report
In this review article, the authors mainly discussed polymeric membrane with lignocellulosic biomass for gas separation. The overall flow of the manuscript is good and a major revision is needed before publication.
- Line 55-60, the authors discussed policies related to greenhouse gas emissions. This part could be shortened and incorporated into previous paragraphs.
- In addition to Table 1, the authors could also discuss the energy cost of different natural gas purification techniques.
- In Table 4, the authors may include more polymer materials for CO2/CH4.
- I’d recommend re-draw Figures 9, 10, and 11.
- Please makes sure the Figure and Table numbers are correct.
- Also, please make sure all significant figures are consistent throughout the paper.
Author Response
The point-by-point rebuttal is attached.

Reviewer 2 Report
That manuscript carries an important insight on the lignocellulose-based MMM for gas separation. This is a subject that is rarely discussed or exploited, thus it carries a great importance to bring forward by Membranes journal to its vast readership.
However, the manuscript is lacking of continuation or information flow and missing is a more detailed discussion on the available MMM systems, among other. In my opinion, the manuscript should be published after the major revision is made.
Below are my comments/suggestions:
- Overall:
- The manuscript is lacking continuation or information flow from paragraph to paragraph and also from section to section. Information rearrangement is needed. Sec. 1, Sec. 2, and Sec. 3 are general information on gas separation membranes and the lignocellulose biomass, respectively. This information is important before reaching the focus on this review paper – which is Sec. 4. However, these sections lack interconnection and seem to be independent of each other. Need to improve.
- What is missing is a more detailed discussion on the available MMM systems using lignocellulose-based fillers. What is the main findings and why it worked (or failed). What is the main gas transport mechanism to have occurred in this hybrid system? In my opinion, this discussion will improve the overall overview of the lignocellulose-based MMMs.
- Also, please standardize the tables.
- Sec, 2
- Line 71-73. The sentence is not complete and it seems like some info is missing.
- Line 79. ‘this field’ refers to what?
- Table 2 is not needed. The kinetic diameters can be included in the text, and the critical temperature is not discussed anyway.
- Line 91. ‘’… inorganic filler which may pose a better interaction with any of the gas components ‘’. In my opinion, the statement is wrong as not all inorganic filler added to have interaction with the permeating gases. Some fillers are to create non-reactive pathways.
- Line 113-115. Hanging sentences.
- Fig. 2. The detail is not complete. Example. Fabrication of membrane (e.g., by..?) and removal of moisture during membrane?
- Line 126. Gas-gas fluid phase? Please clarify this. Do the authors mean the gas-fluid phase as in condensate or gas-gas?
- Line 139-140. The sizes are in regards to the fiber size and not the module. Please reword the sentence.
- Line 153-154. Unclear sentence.
- Line 184-185. Please consider to include the CO2/CH4 newly redefined upper bound in 2019.
- Table 4: Please place the reference accordingly to the data.
- Line 210. Ref. 99?
- Sec. 3
- Line 221. What are those values?
- Line 222-223. I suggest mentioning the ‘major complex components’ so that the sentence will be more complete.
- Line 237. Tress?
- Fig. 6. To increase the size and exclude the ‘used with permission’ disclaimer. The adaptation citation is sufficient.
- In my opinion, Fig. 8 is not needed.
- Fig. 9, Fig. 10 are not clear.
- Table referencing is wrong in this section forward.
- Line 309-310 – what is the main outcome of the study by Xu Li?
- Sec. 4
- Please standardize the tables. Refer to the published papers in Membranes, for example.
- Table 8: MMMs performances for CO2 separation based on filler effects’ is not discussed at all.
- Line 24: Please explain why the CO2 permeability increment is more prominent than CH4?
- Line 362-363. Please explain why the permeability and selectivity decrease simultaneously? What is the non-ideal/defective MMM morphology could have occurred here? Also, can the authors further discuss ‘this can be resolved during doping preparation.’
- Table 10: which polymer is this?
- Line 374-375. What do the authors mean by ‘… are not in line with established trends from the Robeson upper bound curve..’. In my opinion, what you are discussing is the permeability-selectivity trade-off, and if the data is plotted in the double algorithm permeability-selectivity graph – the trend is perfectly fitted to the trade-off behaviour.
- Line 387. 187 Barrer is for which gas?
- Section 4 mainly discusses the perspective of CO2 separation. In my opinion, the prospective (the future) is not discussed thoroughly.
Author Response

(The authors gave the same response as above.)

Round 2
Reviewer 1 Report
The authors have addressed all of my comments and the manuscript is in good shape now.
Author Response
Thank you and we appreciate your satisfaction with our revised manuscript.

Reviewer 2 Report
The authors have improved the manuscript and I thank you for taking my input into consideration. It has a smooth flow and easy to read. My suggestion is that this paper should be published after a minor revision.
The several (very) minor issues are.
- Minor formatting errors:
- Delete line 87
- Missing space between the last work and citation, e.g., word[x]. Occurs in several places.
- Sec. 2:
- Confusing sentences, please consider to reword or restructure:
- Line 77-78. ‘The differences in the sizes of gas particles favor the minimum size difference which accounts for an effective membrane gas separation.’
- Line 81-83: ‘These values indicate the influence of the spherical size of the gas molecules which leads to the possibility of collision between a gas molecule and another molecule’. How does the gas kinetic diameter sizes indicates their collision tendency.
- Lone 140-141: ‘The transport properties of the mechanism of the gas-fluid system are also important’. What do the authors mean here? The gas transport mechanism or?
- Line 165-166: ‘The major diffusion mechanisms are the pores Knudsen diffusion, molecular sieving, and dense membranes solution diffusion’. Maybe change to ‘The major transport mechanisms are Knudsen diffusion and molecular sieving for pored membranes, and solution diffusion for dense membranes’.
- Confusing sentences, please consider to reword or restructure:
- Line 196. ‘the upper bound lines from Robeson’ to just ‘upper bounds’. And it's no longer referred to as Robeson upper bounds as current updates were done by other people.
- Sec. 3:
- Line 235. ‘An annual estimation of global agricultural waste materials produced in 2017 was about 235 37,522,440,479 kilograms while lignocellulosic biomass tops the list’. Those values are for what types of materials and why this values are important to be mentioned?
- Fig. 6, Fig. 8 and Fig. 9 are not clear. I suggest the authors to redraw themselves if the obtained figures from references are not in good quality.
- Typing error in line 275. ‘….transformed ( [130].’
- Sec 4:
- Line 370-373. The font is different.
- Line 410-413: ‘In general, the simultaneous decrease in permeability and selectivity can be attributed to polar and nonpolar gases being separated which is one of the expected setbacks from the separation of gases with close molecular diameters.’ What the authors mean is the competitive sorption – that favour CO2 due to its polarity.
- Table 9 – missing permeability unit.
Author Response
Thank you. All the minor corrections raised have been humbly accepted and corrected on the revised manuscript and can be found on the rebuttal for your reference.
This manuscript is a resubmission of an earlier submission. The following is a list of the peer review reports and author responses from that submission.
Round 1
Reviewer 1 Report
- As suggested by the title and abstract, this review manuscript aimed to review CO2 separation using mixed matrix membranes containing lignocellulosic biomass with focus on the compatibility mechanism. In reality, the manuscript provide and extended introduction on the techniques for CO2 separation (section 1) with focus on membrane separation (section 2) followed by another extended discussion on lignocellulose, its extraction, and pretreatment (section 3). The last section of the manuscript, prospective, discusses the application of lignocellulose in nanocomposites and mixed matrix membranes (not based on lignocellulose) for CO2 The entire manuscript cites 6 references (out of 213 cited references) discusses MMM for CO2 separation and there is not any discussion on lignocellulose MMM for CO2 separation or review of any related references, if any exists. I therefore, find the structure and content of the manuscript to be hetereogenous and not consistent with the title, abstract, and conclusions. For example, in abstract states “In general, the present study has shown that the future membranes generations could produce high permeability and selectivity of CO2 separation using MMMs with less crystalline pretreated lignocellulosic biomasses with accessible hydroxyl group radicals.” What are the basis and evidences that support such statement? The abstract also stated “These shortcomings can be surpassed to improve lignocellulosic biomasses applications by using the proposed pretreatment methods, such as physical and chemical, before incorporation into a single polymer or copolymers matrix.” This seems to be a hypothesis by the author rather than a conclusion from the reviewed literature.
- Other comments:
- The schematic for deadend and breakthrough membrane modules are swapped
Author Response
|
NO. |
COMMENTS FROM REVIEWER |
CORRECTION MADE |
LINE NO. |
|
1 |
In reality, the manuscript provides an extended introduction on the techniques for CO2 separation (section 1) with focus on membrane separation (section 2) followed by another extended discussion on lignocellulose, its extraction, and pretreatment (section 3) |
Thank you for your comment. The article has been revised to show the linkage between mixed-matrix and lignocellulose materials. Composite (mixed-matrix) membranes are obtained by integrating a filler into a continuous matrix of the polymer. The filler can either be organic or inorganic materials. Moreover, the organic filler can either be synthetic or natural. Lignocellulose materials is an example of naturally occurring multi-component material that has been investigated for use as fillers. This linkage is also shown with additional references and deliberations on top some references already in the manuscript which links between lignocellulosic biomass and CO2 separation.
Newly added references are as follows:
Refs. 33 and 34 are added to establish the application of lignocellulosic biomass as fillers to improve separation performance.
The additions are as follows: “An important class of membrane materials that have received a lot of attention is the mixed matrix membrane material. Composite (mixed-matrix) membranes are obtained when a filler material (usually solid phase) is integrated into a continuous matrix of the polymer [33]. The filler can either be organic or inorganic materials. The main aim is to capitalize on the micropores of inorganic filler which may pose a better interaction with any of the gas component [34]. Thus, by incorporating an organic or inorganic filler into a continuous polymer matrix, researchers can exploit the characteristic synergy that results from the nature of the microstructures of both materials”.
The continuous polymer matrix and the organic fillers (which can either be synthetic and natural) are (glassy and rubbery) polymers of varying properties (Table 3).
Refs. 221, 222, and 223 are added to buttress permselectivity improvement by incorporating lignocellulosic biomass loading.
The additions are as follows:
Wu et al. [221] worked on CO2 separation by incorporating bio-cellulose nanofiber (additive) into polymeric membranes. A homogeneous composite membrane solution was obtained by speed coating the additive using the spin coating method and oven drying of the membranes for half an hour at 105 0C. The results showed that polymeric solution concentration increment resulted in increasing the thickness of the membranes. The performance (permeability and selectivity) for 3 wt% loading of the additive was chosen as the best.
Zhang et al. [222] reported 139 Barrer CO2 permeability from cellulose nanofibrils incorporated composite membranes adopting chemical pretreatments to enhance the chemical bonding for high performance.
Venturi et al. [223] used a combination of mechanical and enzymatic pretreatments techniques to prepare nano fibrillated cellulose and incorporated into polyninylamine (PVAm) to fabricate composite membranes. 135 and 187 Barrer was reported for CO2/CH4 ideal selectivity and permeability, respectively. |
Line 88-97 Line 384-395
|
|
2
|
The last section of the manuscript, prospective, discusses the application of lignocellulose in nanocomposites and mixed matrix membranes (not based on lignocellulose) for CO2. The entire manuscript cites 6 references (out of 213 cited references) discusses MMM for CO2 separation and there is not any discussion on lignocellulose MMM for CO2 separation or review of any related references, if any exists. I therefore, find the structure and content of the manuscript to be heterogenous and not consistent with the title, abstract, and conclusions. For example, in abstract states “In general, the present study has shown that the future membranes generations could produce high permeability and selectivity of CO2 separation using MMMs with lass crystalline pretreated lignocellulosic biomasses with accessible hydroxyl group radicals.” What are the basis and evidences that support such statement? |
Thank you for your comment. The article is revised and additional reference and deliberations on lignocellulose MMM for CO2 separation has been added to reflect the scope and coverage of the manuscript to be inline with the Title of the manuscript.
|
Line 364-365
|
|
3 |
The abstract also stated “These shortcomings can be surpassed to improve lignocellulosic biomasses applications by using the proposed pretreatment methods, such as physical and chemical, before incorporation into a single polymer or copolymers matrix.” This seems to be a hypothesis by the author rather than a conclusion from the reviewed literature. |
Thank you for your comment. The statement is not a hypothesis of the authors. Thus, new citation from reference is added to support that statement. The new citation of reference was not added in the Abstract, instead it was added in Section 4, where similar statement was mentioned. |
Line 367-375 |
|
4 |
The schematic for deadend and breakthrough membrane modules are swapped |
The reviewer is correct. Thank you for pointing out this mistake. Thus, the schematic figure (Figure 3) has been swapped accordingly. |
Figure 3, Line 152 |

Reviewer 2 Report
This review paper discussed the potential application of cellulosic biomass in separation membranes. I find there are sufficient new information in the manuscript and the paper is well organised. I am happy to endorse its publication, give some changes can be made by the authors.
- The membrane module in Figure 3 seems incorrect.
- It is welcomed to provide more examples of the cellulosic biomass composite membrane. The author should make a strong case why they would like to pursue this area of study.
- The stability of the composite membrane should be commented. I assume the membrane stability can be a problem after compositing?
Author Response
|
NO |
COMMENTS FROM REVIEWER |
CORRECTIONS MADE |
LINE NO |
|
1
|
This review paper discussed the potential application of cellulosic biomass in separation membranes. I find there are sufficient new information in the manuscript and the paper is well organized. I am happy to endorse its publication, give some changes can be made by the authors |
Thank you and the changes are made on the revised manuscript accordingly following the comments and suggestions of the reviewer. |
|
|
2 |
The membrane module in Figure 3 seems incorrect |
The reviewer is correct. Thank you for pointing out this mistake. Thus, the schematic figure (Figure 3) has been swapped accordingly. |
Figure 3, Line 152 |
|
3 |
It is welcome to provide more examples of the cellulosic biomass composite membrane |
Thank you for your comment. The article is revised and additional reference and deliberations on lignocellulose MMM for CO2 separation has been added to enhance to coverage of the manuscript on cellulosic biomass composite membrane.
|
Line 364-365
|
|
4 |
The author should make a strong case why they would like to pursue this area of study |
Thank you for your comment. The justification has been added as written below:
“A synergistic electronically available literature resources search-words like lignocellulose and membranes revealed that most bio-refined works for biofuel, bio-ethanol, etc. are in majority. An organization (ChemPubSoc Europe) which constitutes 16 chemical societies with eminent professors, is the fore-runner in the valorization of lignocellulosic biomass, indicating the need as an alternative feedstock for energy, environmental, and crude-oil substitute applications. This will open up the doors for interdisciplinary collaborations with colleagues working on catalysis, reaction engineering, among others.” |
Line 338-345 |
|
5 |
The stability of the composite membrane should be commented. I assume the membrane stability can be a problem after compositing? |
Thank you for your comment. As suggested, the stability of the composite membrane has been added and deliberated in the revised manuscript as the following statement:
“Since the lignocellulose biomass contents are natural, anti-degradation and stability can be made robust through the pretreatment techniques which will also prevent fouling. Thus, a thorough understanding of the lignification mechanism before incorporation into the polymer matrix is vital.” |
Line 376-379 |

Reviewer 3 Report
This manuscript to highlight mixed matrix membranes bearing lignocellulosic biomass moieties for carbon capture applications. Referee regretfully thinks that considering broad readership in Membranes, authors should have carefully presented/discussed on efficacy or difference in gas separation performances after loading lignocellulosic moieties in comparison to nascent membranes. Table 8 in the present version of this manuscript seems to be limited numbers of reported literatures to be appealed significance of this review accepted to this journal and there also presented no information on nascent membranes. It could be reasonable to manage Table 4 by including gas separation performances of nascent membranes which have been shown in Table 8. In addition, readers would also raise numerous concerns on differences in gas separation performances depending on pre-treatment methodologies with the same kind of lignocellulosic moieties.
Specific other minor comments are below.
- For a clarity, authors should add recently developed high performance polymers as well into Table 4. For example, PIM-1 and thermally rearranged polymers can be good candidates.
- In line with the comment above, there should be also added relevant ref.s to support the logic; https://doi.org/10.1002/pol.20200110
Author Response
|
NO |
COMMENTS FROM REVIEWER 3 |
CORRECTION MADE |
LINE NO |
|
1 |
This manuscript to highlight mixed matrix membranes bearing lignocellulosic biomass moieties for carbon capture applications. |
Thank you. |
|
|
2 |
Referee regretfully thinks that considering broad readership is membranes, authors should have carefully presented/discussed on efficacy or difference in gas separation performances after loading lignocellulosic moieties in comparison to nascent membranes. |
Thank you for your suggestion. Thus, more information on the gas separation performance after loading lignocellulosic moieties has been included and deliberated in the revised manuscript. |
Line 362-363 |
|
3 |
Table 8 in the present version of this manuscript seems to be limited numbers of reported literatures to be appealed significance of this review accepted to the journal. |
As suggested, more references have been included into Table 8. For the reviewer information, Table 8 only specially presenting the performance of different cellulosic biomass-based membrane for CO2 separation. Whereas there are other Tables such as Table 5 on the pretreatments of lignocellulose, Table 6 on the different sources of cellulose, Table 7 on the properties of cellulose membrane, and Table 9 specifically on the affect of loading cellulose into the membrane for CO2 separation. In a nutshell, this review manuscript discussion not only focus on the performance but the entire value-chain of cellulosic biomass-based membrane from the sources, the pretreatment and performance.
|
Line 187, Table 4 Line 313, Table 5 Line 335, Table 6 Line 358, Table 7 Line 361, Table 8 Line 364, Table 9 |
|
4
|
And there also presented no information on nascent membranes. It could be reasonable to manage Table 4 by including gas separation performances of nascent membranes which have been shown in Table 8 |
Thank you for your comment. The nascent membrane performances are added to Table 4 as recommended as a comparison.
|
Line 187 |
|
5 |
In addition, readers would also raise numerous concerns on differences in gas separation performances depending on pre-treatment methodologies with the same kind of lignocellulosic moieties. |
Thank you for your comment. However, the current manuscript focusing specifically on CO2 separation only. Thus, other gases separation is not covered since it is not under the scope of the current manuscripts.
|
|
|
6
|
For a clarity, authors should add recently developed high performance polymers as well into Table 4. For example, PIM-1 and thermally rearranged polymers can be good candidates. In line with the comment above, there should be also added relevant refs. To support the logic; https://doi.org/10.1002/pol.20200110 |
Thank you for your comment. PIM-1, thermally rearranged polymers, and SPEEK have been added as directed.
Furthermore, new reference as suggested by the reviewer has also been added.
|
Line 187 |

Round 2
Reviewer 1 Report
As suggested by the title, abstract and perspective & Prospective section, this review manuscript deals with the use of lignocellulose in mixed matrix membranes for CO2 separation. However, there are only few studies that report the use of lignocellulose in mixed matrix membranes for CO2 separation (references 219 -223]. Regardless of the changes made to the manuscript this fact cannot be changed.
Therefore, the manuscript title, abstract, and prospective & prospective are not consistent with the content of the manuscript that mainly focuses on two general and separate elements on membrane separation and lignocellulose modification.
Reviewer 3 Report
The authors have adequately addressed all comments. Thank you.